# Identification of Candidate Chemosensory Gene Families by Head Transcriptomes Analysis in the Mexican Fruit Fly, *Anastrepha ludens* Loew (Diptera: Tephritidae)

**DOI:** 10.3390/ijms231810531

**Published:** 2022-09-11

**Authors:** Obdulia L. Segura-León, Brenda Torres-Huerta, Alan Rubén Estrada-Pérez, Juan Cibrián-Tovar, Fidel de la Cruz Hernandez-Hernandez, José Luis Cruz-Jaramillo, José Salvador Meza-Hernández, Fabian Sánchez-Galicia

**Affiliations:** 1Entomology and Acarology Program, Colegio de Postgraduados Campus Montecillo, Km. 36.5, Texcoco 56230, Mexico; 2Escuela Superior de Medicina (ESM), Intituto Politécnico Nacional, Casco de Santo Tomas, Miguel Hidalgo, Mexico City 11340, Mexico; 3Department of Infectomics and Molecular Pathogenesis, Center for Research and Advanced Studies (CINVESTAV-IPN), Mexico City 07360, Mexico; 4Bioinformatics and Technologies Department, Solaria Biodata, Col del Valle Centro, Benito Juarez, Mexico City 03100, Mexico; 5Programa Operativo de Moscas, SENASICA-SADER/IICA, Metapa de Dominguez 30860, Mexico; 6Servicio Nacional de Sanidad Inocuidad y Calidad Agroalimentaria, Secretaría de Agricultura y Desarrollo Rural, Cuauhtemoc, Mexico City 06100, Mexico

**Keywords:** sensorial perception, multigene families, olfactory proteins, pest insect

## Abstract

Insect chemosensory systems, such as smell and taste, are mediated by chemosensory receptor and non-receptor protein families. In the last decade, many studies have focused on discovering these families in Tephritidae species of agricultural importance. However, to date, there is no information on the Mexican fruit fly *Anastrepha ludens* Loew, a priority pest of quarantine importance in Mexico and other countries. This work represents the first effort to identify, classify and characterize the six chemosensory gene families by analyzing two head transcriptomes of sexually immature and mature adults of *A. ludens* from laboratory-reared and wild populations, respectively. We identified 120 chemosensory genes encoding 31 Odorant-Binding Proteins (OBPs), 5 Chemosensory Proteins (CSPs), 2 Sensory Neuron Membrane Proteins (SNMPs), 42 Odorant Receptors (ORs), 17 Ionotropic Receptors (IRs), and 23 Gustatory Receptors (GRs). The 120 described chemosensory proteins of the Mexican fruit fly significantly contribute to the genetic databases of insects, particularly dipterans. Except for some OBPs, this work reports for the first time the repertoire of olfactory proteins for one species of the genus *Anastrepha*, which provides a further basis for studying the olfactory system in the family Tephritidae, one of the most important for its economic and social impact worldwide.

## 1. Introduction

The chemical sensory systems such as smell and taste are part of an insect’s extraordinary capacity to find food, hosts, sexual partners, oviposition sites, avoid natural enemies, and other dangers [1]. The discrimination and interpretation of chemical information in the environment begins in the insect’s olfactory and gustatory organs, with a series of nerve impulses leading to a change in the insect behavior. Before reaching olfactory receptor neurons, there are different biochemical interactions called perireceptor events, that involve the transport of the odorant molecules across the sensillar lymph and the activation of specific transmembrane receptors for the transduction of the chemical signal [2].

This robust and sophisticated system includes several gene families encoding chemosensory proteins involved in complex biochemical reactions in various insect tissues [3,4]. Extracellular Odorant-Binding Proteins (OBPs) and Chemosensory Proteins (CSPs) are the first filters in olfactory processing; they transport hydrophobic compounds from the external environment to specific transmembrane receptors. Odorant Receptors (OR), Ionotropic Receptors (IR), and Gustatory Receptors (GR) initiate downstream signaling related to behavioral responses. Finally, the Sensory Neuron Membrane Proteins (SNMP) play a role in mediating ligand-OR interactions [5,6,7].

The demands of modern agriculture have generated several different challenges, such as creating specific and environmentally friendly alternatives for integrated pest management that reduce or displace the application of insecticides in the future [8]. In this sense, the study of chemosensory genes has increased in insects of agricultural importance [9,10]. Understanding the olfactory system of pests and its role in processes such as host colonization and mating are the basis for the potential development of biotechnological tools such as the creation of biosensors, the computational design of behaviorally active chemicals, and their manipulation by gene silencing techniques [11,12,13,14,15,16,17].

Phytophagous flies of the family Tephritidae, commonly called fruit flies, are among the most economically significant insect groups, causing primary fruit and horticultural losses worldwide in tropical and subtropical regions [18,19]. In the Americas, the largest genus of Tephritidae is *Anastrepha* Schiner, with more than 250 species distributed from the southern United States to northern Argentina [20,21]. In Mexico, *A. ludens*, or the “Mexican fruit fly”, is a pest of quarantine importance that represents a major phytosanitary problem in various fruits, particularly citrus and mango [22,23,24]. It caused direct damage due to crop yield losses, increased control costs, and international marketing restrictions [25].

In the last decade, an increasing number of studies have focused on discovering the chemosensory genes of pest tephritid species such as *Bactrocera dorsalis*, *B. minax*, *Zeugodacus cucurbitae*, *Z. tau*, *Ceratitis capitata*, *A. fraterculus*, and *A. obliqua*, to understand the olfactory pathways and their role in fruit flies’ behavior [26,27,28,29,30,31,32]. To date, there is no information on the multigene families of the olfactory system of *A. ludens* [33,34], but new sequencing technologies and advances in model insect communication systems offer the opportunity to understand similar mechanisms in the Mexican fruit fly.

Identifying putative chemosensory genes in *A. ludens* is the first step to exploring crucial gene functions in the communication process. Their knowledge can provide further insight into the evolutionary history of the genus *Anastrepha*, related to the search behavior, host selection, and mating in recently diverged species with a limited number of distinctive morphological and genetic characters [27,35]. In this study, we identified, analyzed, and characterized the six chemosensory gene families of *A. ludens* by analyzing two head transcriptomes of sexually immature and mature adults from laboratory-reared and wild populations, respectively.

## 2. Results

### 2.1. Transcriptome Assembly and Annotation

As a result of sequencing five Illumina HiSeq2000 libraries of sexually immature male and female heads of *A. ludens*, we obtained an average of 6.82 million clean reads, 92.37% with a Phred quality score Q20. The clean reads were assembled by de novo into 24,368 transcripts, and after removing redundancies and isoforms, 21,103 unigenes were obtained with a mean length of 359 bp and an N50 of 880 (Appendix A). On the other hand, with the six paired libraries generated with the BGISEQ-500 platform from sexually mature male heads of *A. ludens*, an average of 47.94 million clean reads were obtained with a length of 99 bp, and more than 97% of reads had a Phred quality score Q20 and low duplication rates. The de novo assembly resulted in 269,924 transcripts and after removing redundancies, we obtained 158,693 unigenes with a mean length of 358 bp and N50 of 1245 (Appendix A).

We used Gene Ontology (GO), UniProtKB, and InterPro databases (accessed on 1 October 2021) to annotate unigenes obtained from *A. ludens* head transcriptomes from Illumina HiSeq2000 and BGISEQ-500 platforms. A total of 13,723 (65.02%) unigenes obtained with Illumina (Figure 1a) and 39,696 (25.01%) unigenes generated with BGI were recorded in at least one database (Figure 1). In both cases, of the total number of annotated unigenes, more than 80% were recorded in the Insecta-UniprotKB, more than 50% in the InterPro, and less than 40% in the GO databases (accessed on 1 October 2021). In both *A. ludens* transcriptomes, more than 90% of the genes had hits with sequences from the family Tephritidae, mostly with sequences from *B. dorsalis*, *B. tryoni*, *C. capitata*, *B. latifrons*, and *Z. cucurbitae*, and only 1.05% matched sequences from seven species of the genus *Anastrepha*. In addition, more than 50% of the genes had E-value lower than 1 × 10^−50^ and similarities higher than 70% (Figure 1).

Transcriptomes generated with Illumina HiSeq2000 and BGISEQ-500 were annotated in the three main GO categories. The molecular function category was the most representative, and genes enriched with more than 50% and 30% were binding and catalytic activity. Cellular processes and metabolic terms were the most represented subcategories in biological processes. In contrast, we classified more than 50% of unigenes in a cellular component in the subcategories of cell part, cell, and membrane (Appendix A).

### 2.2. Mining of Chemosensory Genes

We mined the protein-coding transcripts with sequence similarities to the six chemosensory families using the Insecta base of UniProtKB and identified a total of 224 transcripts homologous to communication genes; 95 belong to non-receptor families, and 132 to receptor families. We obtained the open reading frames (ORFs) of the 224 transcripts and performed Blastp analysis with protein domain searches as retention criteria to maximize sensitivity and obtain functionally significant sequences. After functional analysis and elimination of redundant sequences, the number of filtered genes was reduced to 120 transcripts encoding 31 AludOBPs, 5 AludCSPs, 2 AludSNMPs, 42 AludORs, 23 AludGRs, and 17 AludIRs.

### 2.3. Odorant Binding Proteins (OBPs)

We identified thirty-one candidate OBP transcripts encoding proteins ranging in length from 102 to 306 amino acids. Of the 31 sequences, twenty-one had complete ORFs, seven partial ORFs with truncation in the 5′ region, three internal partials, and 19 AludOBPs had signal peptides. All proteins were functionally annotated by searching for domains within the insect pheromone superfamily/odor-binding proteins (SSF47565) and PBP/GOBP family (PF01395). All 31 OBPs had homologs with odor proteins of the family Tephritidae, 19 AludOBPs had similarities to *A. obliqua* and *A. fraterculus* with values mostly higher than 90%, and sequences with similarities to OBPs from *B. dorsalis*, *C. capitata*, and *Z. tau* were also recorded (Appendix A).

The classification of the AludOBPs was performed based on the number and location of the conserved cysteines; we grouped 31 proteins into classical, Minus-C, Plus-C, and dimer subfamilies (Table 1). The subfamily with the highest number of AludOBPs (17) was classical, which presented the general pattern of the six conserved cysteines and some with additional cysteines upstream of C1 (Appendix A). We also identified eight AludOBPs Minus-C, which only conserved four cysteines and had a different number of residues between them. AludOBP49a1, AludOBP49a2, AludOBP50a, and AludOBP50e were classified within the Plus-C subfamily, which presented additional cysteines at the N-terminal end, two or three additional cysteines downstream of C6 and a conserved proline after the seventh cysteine. AludOBP83ef and AludOBP83cd presented two six-cysteine motifs classified within the dimeric subfamily (Appendix A).

On the other hand, we constructed a phylogenetic tree by Bayesian Inference of candidate OBPs of *A. ludens* with proteins from seven Tephritidae species and the model organism *Drosophila melanogaster* (Figure 2). Most of the AludOBPs clustered into different subclades with their closest homologs from *A. fraterculus* and *A. obliqua* with 100% Bayesian posterior probabilities (BPP), related to proteins from other fruit flies species and *D. melanogaster*, showing highly supported terminal clusters of proteins with specific lineage expansions.

Four groups representing the subfamilies identified in the AludOBPs were defined in the phylogenetic tree; the Minus-C and dimer members were more closely related and clustered within the same clade with high supports (>90% BPP). Most Minus-C in Tephritidae are classified as OBP99a; however, in our analysis, several of them are related to OBPs from *D. melanogaster* OBP44a, OBP83g, OBP99b, OBP8a, and OBP99c, suggesting a classification that has reduced the representation of members of this subfamily and their diversification into other fruit fly genera.

The analysis grouped the classical family into three major clades; the group containing the proteins AludOBP73a and AludOBP59a was related to members of the Plus-C subfamily, with a higher number of residues (≈171) between C1 and C2. On the other hand, the phylogeny clustered two genes encoding OBP19a (AludOBP19a1 and AludOBP19a2) with DmelOBP19a and the OBP56h were divided into two groups within the same clade, each related to different DmelOBPs (100% BPP). Likewise, within the classical AludOBPs, we identified proteins related to *D. melanogaster* proteins as AludOBP83a and *B. dorsalis* and *Z. tau* proteins as AludOBP73a and AludOBP83a (Figure 2).

### 2.4. Chemosensory Proteins (CSPs)

We identified five candidate *CSP* transcripts from *A. ludens*, four encode proteins with complete ORFs and lengths between 120 and 149 aa, while AludCSP3-1 had a partial ORF in the 3′ region and length of 51 aa. All AludCSPs had annotations by conserved domains within the chemosensory protein superfamily CSP2 (SSF100910) and insect odorant-binding domains such as A10/Ejaculate Bulb Specific Protein (PF03392), related to chemosensory proteins (PTHR11257). The five AludCSPs had homologs with odor proteins of fruit flies *B. minax*, *B. dorsalis*, and *Z. tau* with identities higher than 79% (Appendix A).

The predicted AludCSPs proteins have all four conserved cysteine residues (C1-X_6_-C2-X_18_-C3-X_2_-C4). The phylogenetic analysis grouped the AludCSPs in two major clades with 100% BPP. The five AludCSPs were dispersed into five well-defined subgroups related to their orthologues from tephritid species, and except for the clade of AludCSP3-2, all CSPs from tephritid were clustered with one or two DmelCSPs with BPP > 90% (Figure 3). Sixteen of the tephritid and three *D. melanogaster* sequences homologous to AludCSPs, are not characterized in the NCBI non-redundant database (accessed on 1 August 2022) but have identity percentages >80% with AludCSPs, and present the characteristic protein domains of the CSP family (A10/PEBIII/OSD).

### 2.5. Sensory Neuron Membrane Proteins (SNMPs)

Two SNMP transcripts were identified in *A. ludens* head transcriptomes encoding 509 and 630 residue proteins with complete ORF and two transmembrane regions at the C- and N-terminal ends (Appendix A). The AludSNMPs had homologs with SNMP1 and SNMP2 from *C. capitata* and *B. dorsalis* with an E-value of zero and identities higher than 90% (Appendix A). The annotations based on conserved domains correspond to the CD36 family (PF01130). In the phylogenetic analysis of AludSNMPs with proteins from seven dipteran species (Figure 4), SNMPs were separated into SNMP2 and SNMP1 with 100% BPP. The AludSNMPs were more closely related to *C. capitata* proteins within each group. Furthermore, SNMP1 split into two groups, and the one identified for *A. ludens* grouped within SNMP1b.

### 2.6. Odorant Receptors (ORs)

We identified 42 *OR* genes, 35-recorded complete ORFs encoding proteins with lengths above 357 residues, and seven genes had partial ORFs in the 5′ or 3′ regions; however, only three of them had lengths less than 300 amino acids. We annotated all AludORs within the insect odorant receptor family (PF0249; Pfam: PF02949) and odorant receptor subfamilies (PANTHER: PTHR11857). AludORs had five to seven transmembrane domains, except for AludOR2a, AludOR7a6, AludOR71a1, AludOR85d, and AludOR88a; however, only three of them have partial sequences (Appendix A).

We performed two homology analyses for receptor annotation with the NCBI and UniprotKB-Insecta non-redundant protein databases (accessed on 1 February 2022). In both cases, all AludORs had homologs with dipteran proteins. With the UniprotKB base, almost 90% of the AludORs matched with proteins from seven tephritid species with identities of 30–96% (Appendix A). In contrast, with the non-redundant protein base, more than 50% of the AludORs had homologs with putative odorant receptors from *R. pomonella*, *B. oleae*, *C. capitata*, and *B. tryoni*, with higher similarity percentages (>70%) (Appendix A). One of the AludOR proteins, AludORCO, shares 96.19% and 95.77% of identity with a co-receptor ORCO/OR83b from *Procecidochares utilis* and *B. tryoni*, respectively.

To gain insight into the evolutionary relationships of the 42 AludORs, we performed a phylogenetic analysis by Bayesian Inference using a dataset of 283 odorant receptor sequences from eight tephritid species and the model fly *D. melanogaster* (Figure 5). ORs clustered into three major clades with 90% BPP; AludORs dispersed into subgroups related to their orthologues from tephritid species, most closely related to *Rhagoletis* species. AludORCO is clustered in a highly conserved clade with other olfactory co-receptors (OR83b) closely related to OR83a.

AludORs were organized with homologous sequences into 36 orthogroups with lineage-specific expansions; 29 consisted of well-defined single clusters (no duplications), of which 26 were related to homologous *D. melanogaster* receptors with >90% BPP. Three gene clusters (OR67c1, OR42a, and OR43a) did not show clustering with the model fly but with genes from at least six tephritid species (Figure 5). The remaining clusters included specific duplications such as OR71a, OR69a, and OR67d and specific expansions such as OR7a and OR74a (Figure 5).

### 2.7. Gustatory Receptors (GRs)

Twenty-three transcripts similar to *GR* were identified, encoding proteins with an average length of 350 residues. Ten had complete ORFs with five to eight transmembrane domains, whereas three to seven transmembrane domains were recorded in partial transcripts (Appendix A). The AludGRs had homologs with UniprotKB-base dipteran sequences, 20 AludGRs with similarity to *B. correcta*, *B. dorsalis*, *B. latifrons*, *C. capitata*, and *Procecidochares utilis* with identities mostly higher than 70% and three with proteins from *D. ananassae*, *D. melanogaster*, and *D. mojavensis* with similarities less than 60%.

AludGRs had similarities to different taste receptors of *B. oleae*, *B. tryoni*, and *R. pomonella*, more than 60% with an E-value of zero and identity percentages higher than 55% (Appendix A). InterPro analysis classified *A. ludens* GRs into the invertebrate gustatory receptor (PANTHER: PTHR21143) and chemosensory receptor (Pfam: PF08395) families. The phylogenetic analysis clustered the 23 AludGRs into three clades clustered with homologous sequences from seven tephritid species and the model species *D. melanogaster* with 100% BPP, most closely related to *R. pomonella* GRs, and other groups to *C. capitata* and *B. latifrons* (Figure 6).

### 2.8. Ionotropic Receptors (IRs)

We identified seventeen *iGluRs/IRs*-related transcripts in *A. ludens* head transcriptomes, ten with complete ORFs encoding proteins of 626–952 residues and seven partial ORFs with lengths above 180 residues (Appendix A). All AludIRs had transmembrane domains and annotation within the ionotropic receptor family (PANTHER: PTHR42643) and ion channel ligand-gated L-glutamate and glycine binding site (Pfam: PF10613). AludIRs had homology with proteins from *B. correcta*, *B. dorsalis*, *B. latifrons*, *C. capitata*, *Z. tau*, and *Z. cucurbitae* with identities of higher than 60%, except AludIR7c and AludIR31a that had similarities of 40% and 49% with IRs from *D. lebanonensis* and *Musca domestica* (Appendix A).

On the other hand, homology analyses with the NCBI non-redundant protein database (accessed on 1 February 2022) yielded different results, AludiGluRs/IRs had homologs only with proteins from the Tephritidae family with an E-value of zero and similarities higher than 70%, except for AludIR31a, and in contrast to UniProtKB, most of those proteins were not annotated (Appendix A). Phylogenetic analysis using Bayesian Inference corroborates the annotation of AludiGluRs/IRs with previously described homologous receptors from six tephritid species and *D. melanogaster* (Figure 7).

Three AludIRs members of the divergent IR clade (IR7c, 56c, and 100a) were identified; the group of antennal IRs was the largest, with ten members. The analysis clustered AludIR40a, AludIR41a, AludIR76c, and AludIR92a with their orthologues within the divergent clade. In addition, we identified a specific group for the antennal IRs AludIR31a, AludIR75a, AludIR75a2, and AludIR75d, while the receptors AludIR21a and AludIR93a were related to the putative co-receptors IR25a/IR8a and the non-NMD iGluRs AludGluIID and AludKaiR1D.

## 3. Discussion

The study of molecular communication in Tephritidae has boomed in the last fifteen years, for the potential benefits that understanding the mechanisms of chemosensory perception offers in developing strategies for the management of these insects [32]. Currently, the UniprotKB protein database (accessed on 1 February 2022) contains 1558 chemosensory proteins belonging to thirteen species of seven tephritid genera. Still, only 3% correspond to sequences from two species of the genus *Anastrepha* (*A. fraterculus* and *A. obliqua*). This work is the first report on the assembly and analysis of two transcriptomes of adult females and males of the Mexican fruit fly *A. ludens* and identifying 120 putative genes encoding proteins of the six chemosensory receptors and non-receptor families.

We performed *A. ludens* head transcriptomes with two different sequencing platforms, the number of unigenes obtained was higher with BGISEQ-500, but in general, both transcriptomes had a low percentage of annotated transcripts with respect to the total number of assembled unigenes. The differences in the number of unigenes obtained and annotation rates may be due to the specimen collection conditions and the number of libraries assembled with each platform. Likewise, other work has suggested that the low percentage of transcripts annotated in insect tissues may be due to genes that are not homologous to those deposited in the databases, high sequence divergence, or partial transcripts with no match [36,37].

Even with these differences, in both transcriptomes, more than 80% of unigenes translations shared significant similarities with entries from the UniprotKB database (accessed on 1 November 2021) of the order Diptera, followed by InterPro (≈50%), and less than 40% of transcripts fell into all three categories of GO gene ontology terms. The most abundant GO functional groups in both transcriptomes showed similar frequency in binding and catalytic activity, cellular, and metabolic process. Comparable to that reported in antennal and head transcriptomes of other dipteran species and insect orders, the similarity in the annotations of the two data sets indicates some level of conservation in gene expression in sensory tissues [36,38,39,40].

Different olfactory-related genes have been explored in tephritid species as potential candidates for developing biotechnological applications for pest management [26,41,42,43,44,45,46,47]. OBPs are small water-soluble proteins that transport odors to receptors in olfactory neurons and play a fundamental role in natural and sexual selection [48,49]. There are more than 15,000 amino acid sequences of insect OBPs in the NCBI database (accessed on 1 February 2022); however, for the genus *Anastrepha*, there is only information on 47 OBPs [27,50,51].

We identified 31 *A. ludens* genes encoding OBPs, this is higher than reported in head and reproductive tissue transcriptomes of the sister species *A. obliqua* (24) and *A. fraterculus* (23) but similar to OBPs identified in *B. dorsalis* (49), *B. papayae* (35), *B. correcta* (34), *Z. cucurbitae* (33), *Z. tau* (33), and *C. capitata* (34) [31,52,53]. The differences in putative OBPs described for the three *Anastrepha* species may be due to the physiological stage at which the transcriptomes were performed, related to the abundance of the genes of interest and the coverage obtained with the sequencing platforms [27,31]. Although this study increases the number of OBPs described for this genus, further studies in different tissues may increase the number of OBPs of the three *Anastrepha* species.

Comparative genomic analysis of different hexapod species has shown a very dynamic evolution of OBPs, with a wide range of protein lengths with varying profiles of cysteine [54,55,56]. The classification of the AludOBPs was identical to that described for *A. fraterculus* and *A. obliqua*; the largest subfamily was the classical subfamily with 17 proteins, followed by Minus-C (8), Plus-C (4), and we also identified two members of the dimer subfamily [42]. Most AludOBPs clustered with the closest orthologues of *Anastrepha*, supported with *D. melanogaster* OBPs with BPP >90%. Based on the relationships of AludOBPs with *D. melanogaster* orthologues, we made a name adjustment to make the proteins comparable with other studies.

In contrast to Campanini and Brito (2016) [27], we did not observe a basal division into four clades corresponding to each subfamily; the phylogenetic analyses clustered the members of the classical subfamily in three clades, one of them related to the Plus-C subfamily. As in other studies, OBPs of the dimer and Minus-C subfamily seem to have had independent origins, as they clustered in an external clade with 100% BPP supports [49]. In contrast to *A. fraterculus* and *A. obliqua*, we identified six additional AludOBPs in the classical subfamily (AludOBP28a, AludOBP73a, AludOBP84a, AludOBP19a1, AludOBP83a, and AludOBP19d2), all related to orthologues of *D. melanogaster* and tephritid except AludOBP19b-like, which was related to the OBP19b group and shared a 37.04% similarity to AludOBP19b.

It is common in OBPs to find homologous copies of a gene in the same species derived by the duplication process [55,56]. Campanini et al. (2017) [51] mentioned that the OBP56h group in different tephritid presents a duplication event and reported positive selection between AfraOBP56h-1/AoblOBP56h-1 and AfraOBP56h-2/AoblOBP56h-2. Initially, we identified two homologous members to OBP56h in *A. ludens* OBPs; however, their identity was less than 20%, so we corroborated the annotation against the *D. melanogaster* reference protein and observed identities of both proteins with OBP56h and OBP56g. Likewise, phylogenetic analysis showed the separation of tephritid OBP56h into two clades, each related to DmelOBP56g and DmelOBP56h with 100% BPP, suggesting two closely related sister groups with specific diversification.

On the other hand, OBP73a, which was related to the OBP59a, was characterized by having the presence of a large number of residues (>100) between C1 and C2, which had not been described in other *Anastrepha* transcriptomes [49,57,58,59]. Interestingly both proteins clustered within the same clade with the Plus-C subfamily; however, their unusual lengths could generate branch attraction, which may not reveal a true evolutionary history [60]. As in *A. obliqua*, we identified two copies of the Plus-C OBP49a proteins; the gene encoding this protein was not identified in *A. fraterculus*, suggesting diversification after the separation of the species [27]. The high divergence at the amino acid level and its identification in the Mexican fruit fly could indicate that this gene is present but not detected in *A. fraterculus*, probably because of low expression levels during the collection stages.

The CSP gene family is highly conserved in insects; comparative analyses of the genomes of different insect orders reported that, in contrast to OBPs, the repertoire of CSPs is markedly reduced from three members in the genome of *D. melanogaster* to 20 in *Tribolium castaneum* [55,61,62,63,64,65,66]. For Tephritidae, there are few published papers for these non-receptor families; in general, three to five members of CSPs have been reported in *B. dorsalis* and *B. minax* [29,30,31,67,68].

We obtained equivalent results in the transcriptome of *A. ludens*, with five members recorded for CSPs. Similar to Cheng et al. (2020) [31], we identified two genes that encode CSP3. According to the phylogenetic tree, AludCSP3-1 and AludCSP3-2 belong to two subgroups with a specific diversification, related to orthologs of different tephritid species, of which the CSPs of the database for *B. dorsalis*, *B. tryoni*, *Z. tau*, and *Z. cucurbitae*, also have two different genes encoding CSP3. However, only the cluster AludCSP3-1 is related to an orthologue with *D. melanogaster* A10/OSD.

Although this family is more conserved than OBPs and is more reduced in Diptera, knowledge about it is evolution and function in Tephritidae is limited to a few species [29,69,70], and most of the sequences for fruit flies are not reported and characterized. Even for *D. melanogaster*, three sequences homologous to the AludCSPs were not classified within the CSPs, even though they present the characteristic protein domains of this family. Therefore, it is necessary to carry out comparative studies of the information deposited in the world databases to unify the classification of CSP members in Tephritidae and define the phylogenetic relationships with the functional characterization.

On the other hand, we identified two genes encoding SNMPs in the head transcriptomes, both of which presented the characteristics of this family, including the two transmembrane regions and six conserved Cys residues [5,70]. Phylogenetic analysis corroborated the division of AludSNMPs and their orthologs into SNMP1 and SNMP2 subgroups with 100% BPP support [71,72,73]. SNMP1 splits into SNMP1a and SNMP1b subtypes, in four species of *Bactrocera* and two species of *Zeugodacus*; however, in the transcriptome of *A. ludens*, we only identified SNMP1b. In addition, we identified a gene encoding SNMP2, also described in antennae and non-olfactory tissues, with high expression in legs in *B. minax* [53,73].

Three gene families encode chemosensory receptors; insect ORs detect volatile chemical information and are very different from mammalian GPCR with an inverse heptahelical topology; the GRs that are part of the gustatory system perceive a broad spectrum of ligands, and those related to ionotropic glutamate receptors are called ionotropic receptors (IRs) [4]. In recent years, knowledge of chemosensory receptors in agriculturally important tephritid has increased; however, there are no reports or information in genetic databases (accessed on 1 September 2022) of these receptors in the genus *Anastrepha* [74,75,76]. This work reports for the first time the identification and analysis of eighty-two genes encoding the three-receptor families in the Mexican fruit fly.

The number of putative ORs we identified in *A. ludens* (42) is similar to that reported in *B. dorsalis* (43) and higher than other *Bactrocera* and *Zeugodacus* species that vary in the range of 39 to 41 genes. This result suggests that the ORs not only show conservation in gene number within the Dacinae group but is comparable to other tephritid subfamilies [30,77]. As in other insects, A. ludens has one ORCO gene that shares a high similarity (>90%) to co-receptors of the eight tephritid species analyzed. ORCO is coexpressed with a specific OR in nearly all olfactory neurons [11] and plays an essential role in response to common odors, including esters, aldehydes, ketones, aromatics, and terpenes; in the male oriental fruit fly B. dorsalis, Orco participates in the taxis to methyl eugenol [78].

*A. ludens* has expansions of AludOR7a/AludOR74a and specific duplications of AludOR71a, AludOR69a, and AludOR67d. Different studies report OR7a and OR74a as positive signaling receptors to aggregation compounds such as 9-tricosene and 1-nonanol/6-oxo-1-nonanol. Diversification of these genes in tephritid may enhance the perception of specific odorants or the combination of similar odorants, which may be related to species differences in the detection of host volatiles and pheromones [32,53,79,80].

Although the number of *A. ludens* ORs is identical to that of another Tephritidae, we found differences in the previously reported homologous. The putative receptors AludOR30a-like1, AludOR30a-like2, AludOR59a-like, and AludOR63a-like, only clustered with receptors from *R. pomonella*, *R. zephyria* derived from an annotated genomic sequence using the gene prediction method, whereas AludOR59b-like and AludOR5 clustered with two orthologues from *B. minax* and *C. capitata*. Although the six receptors presented ambiguous classification and did not cluster with receptors from other tephritid species or *D. melanogaster*, all, except AludOR59a-like and AludOR30a-like1, present complete ORFs and five to seven predicted transmembrane regions. Furthermore, the phylogenetic analysis clustered AludOR30a-like1 and AludOR30a-like2 in the same clade related to OR49b and present an identity of 30.77%.

On the other hand, GRs repertoires vary depending on the insects’ order, from 10 genes in the honeybee *Apis melifera* to 68 and 76 genes in *D. melanogaster* and the mosquito *Anopheles gambiae* [81,82,83,84]. In this work, we identified 24 GRs in the transcriptomes of *A. ludens*, which is higher than that reported in Tephritidae, from six GRs in *Z. tau*, to 15 in *B. minax*. Several studies have shown associations in the number of GRs and host range, where specialist insects conserve a lower number of GRs than generalists [74,85,86]. However, similar to the Mexican fly, the reported Tephritidae species are characterized by a broad host range [23]. Given that GRs present a high sequence divergence, making their identification difficult, it is necessary to perform comparative genome analyses to corroborate the GRs’ repertoire [32,74].

Phylogenetic analysis grouped the twenty-three AludGRs and their orthologues into three clades representing the different lineages conserved in insects with specific putative functions; AludGR63a, AludGR21a, and AludGR22 belong to CO_2_ receptors. The most conserved in insects. GR22, homologous to GR21a, was found expanded in Tephritidae, which could enhance their olfactory sensitivity to CO_2_ [53,87,88]. Seven AludGRs (AludGR64a-f and AludGR5a) were classified within the sugar receptors, and we identified a duplication of the AludGR64b gene related to the Dmel64b protein. The remaining 13 AludGRs group into the bitter/aversive receptor clade, representing a high proportion within this family with specific expansions. Some of them, such as AludGR98a, AludGR98b, AludGR2a, and AludGR22e, are only related to *R. pomonella* and *D. melanogaster* GRs [89,90,91].

We identified 17 putative IR/iGluR-encoding genes in *A. ludens*; this is lower than reported in six fruit fly species with an average of 23 IRs and *D. melanogaster* with 66 IRs [30,53,92]. Based on phylogenetic analysis and Croset et al., 2010 [93], the 17 AludIRs were classified into the subfamilies divergent IRs, antennal IRs, non-NMDA (N-Methyl-D-aspartic acid), iGluRs, and two putative co-receptors. Although the antennal subfamily represents only a fraction of the IR repertoire in most insects, in *A. ludens* transcriptomes, it was the most representative group with ten members. All AludIRs formed orthologous groups with *D. melanogaster* receptors, except for IR75a, which presents an expansion in tephritid and could enhance their ability to perceive fermented fruits [30,53,94].

Croset et al. (2010) [93] mentioned that dipteran insects have broad expansions of divergent IRs; however, in *A. ludens*, we only identified three members of this subfamily (AludIR7c, AludIR56c, and AludIR100a), which were closely related to three clusters of antennal IRs with 90% BPP. Several evolutionary and gene expression studies report the IR25a cluster as a putative co-receptor, analogous to the heteromeric assembly of iGluR subunits into functional complexes [95]. In addition, IR25 appears to be an atypical member that, together with its homolog IR8a, shows high conservation with the primary sequence to iGluRs. The above is consistent with the phylogenetic relationship of AludIR25a and AludIR8a, located within the iGluR family of non-NMDA receptors, suggesting that these IRs are among the ancestral groups [92,93,96].

In conclusion, this work represents the first effort to identify, classify and functionally characterize the six chemosensory gene families from the analysis of head transcriptomes of male and female adults of *A. ludens* in different physiological stages. The 154 described genes encoding putative proteins play an essential role in the chemosensory perception of the Mexican fruit fly. They are a significant contribution to the genetic databases of insects, particularly dipterans, which will provide a further basis for the study of molecular communication in the Tephritidae family, one of the most important for its economic and social impact worldwide. Moreover, the identification of olfactory genes provides information for the study and development of alternative strategies for the specific management of pests of agricultural importance.

## 4. Materials and Methods

### 4.1. Collection of Biological Material

In 2017, we collected sexually immature adult females and males between four and 11 days of the emergence of *A. ludens* from the laboratory-reared of the Colegio de Postgraduados, Campus Montecillo; heads were dissected and stored in tubes with 600 µL of Thermo Fisher RNAlater™ buffer (Thermo Fisher Scientific, Inc., Lithuania). On the other hand, in 2020, a direct collection of mangoes infested with wild *A. ludens* larvae was carried out in Tapachula, Chiapas. The fruits were placed in cages until the emergence of adults, which were kept in rearing with a 12 L (07:00–19:00 h):12 O photoperiod and fed on sweetened water: hydrolyzed protein mixture until sexual maturity. Then, sexually mature males of eighteen-day-old were collected at resting (10:00–12:00 a.m.) and calling (6:30–7:00 p.m.) physiology stages; the specie was taxonomically corroborated by the technical experts of the MOSCAMED program, and heads were dissected and preserved in 200 mL of RNAlater™.

### 4.2. Preparation and Sequencing of cDNA Libraries

For the breeding material collected in 2017, we performed four total RNA extractions with the SV Total RNA Isolation System kit (Promega, Madison, WI, USA), two replicates of twenty-five male and female heads (1:1) for each emergence time (4 and 11 days), and we evaluated the extraction quality with a Qubit™ fluorometer (Thermo Fisher Scientific, Inc., Waltham, MA, USA). Four paired libraries were constructed with the TruSeq RNA Sample Preparation Kit V3 (Illumina Inc., San Diego, CA, USA) and sequenced with the MiSeq platform at the Centro Nacional de Referencia Fitosanitaria, SENASICA, Tecámac, State of Mexico. On the other hand, the heads of wild males collected in mango were sent to BGI Hong Kong, China. We performed RNA extraction in three replicates of twenty-five male heads for each physiological stage; the quality and concentration were verified with an Agilent 2100 Bioanalyzer (Agilent Technologies, Inc., Santa Clara, CA, USA) and constructed six paired cDNA libraries with the MGIEasy-RNA Library-Prep kit (MGI Tech Co., Ltd., Hong Kong, China). Sequencing was performed with the BGISEQ-500 platform (BGI Group, Hong Kong, China), with a paired read length of 100 bp and an output of 4 GB of clean data.

### 4.3. Cleaning, Assembly, and Annotation of the De Novo Transcriptome

Ten paired libraries were sequenced with Illumina and BGI platforms and evaluated the quality with FastQC v0.10.1 (Babraham Institute, Cambridge) [97]. The cleanup of adapters and low-quality reads was performed based on the characteristics of each platform with FastP (https://github.com/OpenGene/fastp, accessed on 1 September 2021) [98]. The libraries were assembled de novo based on the sequencing platform with Trinity v2.06 (https://github.com/trinityrnaseq/trinityrnaseq, accessed on 1 September 2021) and a default value of kmer = 25. The quality of the assemblies was evaluated with QUAST v.0.3.0. (https://github.com/ablab/quast, accessed on 1 September 2021) [99]. We recorded the total number of reads generated, length N50, mean length, and the GC% and filtered out redundancies to obtain the final number of unigenes for each assembly. We performed the annotation of the unigenes of the two assemblies by homology analysis with BLASTx [100] against the Insecta database of UniprotKB (3,914,796 sequences) with an E-value of 1 × 10^−7^. We added a conserved protein domain search with InteproScan workflow [87] and obtained the Gene Ontology (GO) terms with HMMER2GO (https://github.com/sestaton/HMMER2GO, accessed on 1 October 2021).

### 4.4. Mining and Functional Analysis of Chemosensory Genes

We filtered the sequences of *A. ludens* chemosensory receptor and non-receptor protein families from the results obtained in BLASTx homology analysis against the UniprotKB base and conserved InterProScan domains. We verified the open reading frames of the sequences with ORFinder (https://www.ncbi.nlm.nih.gov/orffinder/, accessed on 1 November 2021) and performed Blastp analysis against the UniprotKB database (accessed on 1 February 2022), and for the receptors, with the NCBI non-redundant protein base (accessed on 1 February 2022). To predict the protein domains of the receptor proteins (including SNMPs), we used TMHMM 2.0 (DTU Health Tech, Lyngby, Denmark) [101] and SignalP-5.0. (DTU Health Tech, Lyngby, Denmark) [102] to predict the presence of signal peptides in OBPs and CSPs. We evaluated the sequences by multialignment. For OBPs, we generated a classification based on cysteine profiles and obtained the average amino acid identities with the MAFFT percent identity matrix. The final sequences of the six chemosensory protein families were named Alud, followed by the corresponding protein name based on the best match in the Blastp analysis.

### 4.5. Sequence Comparison and Phylogenetic Analysis

We generated six different datasets for each candidate chemosensory families of *A. ludens* and homologous protein sequences from diverse tephritid species (*B. dorsalis*, *B. latifrons*, *B. correcta*, *B. minax*, *B. tryoni*, *B. oleae*, *Z. cucurbitae, Z. tau*, *C. capitata*, *R. pomonella*, *R. zephyria*, *P. utilis, A. obliqua*, and *A. fraterculus*) and the model species *D. melanogaster*. In addition, we included proteins from the mosquito *Anopheles gambiae* for SNMPs analysis (Appendix A). For multiple alignments of OBPs and CSPs, Clustal Omega (UCD Dublin, Dublin, Ireland) [103] was used with default parameters, while for SNMPs and receptors; we made alignments with the iterative method with MAFFT refinement [104]. We visualized the alignments in AliView (Uppsala University, Uppsala, Sweden) [105]) and eliminated the extremes in short or highly divergent length sequences. For each dataset, we obtained the best protein evolutionary model in ModelTest-NG (https://github.com/ddarriba/modeltest, accessed on 1 April 2022) [106] using the AIC and BIC criteria and constructed six phylogenetic trees by Bayesian inference with the Markov chain Monte Carlo method in BEAST v1.10.4 (© BEAST Developers) [107] with a minimum of 15,000,000 generations (Appendix A). We evaluated the MCM output in Tracer v1.7.1 (©BEAST Developers) and incremented the chains for those analyses that required it. We discarded 30% of the initial trees for tree annotation and calculated posterior probability with the remaining trees. Finally, we visualized and edited the consensus trees in iTOL v6.5.8 (European Molecular Biology Laboratory, © EMBL) [108].

### 4.6. Sequence Information

*A. ludens* chemosensory gene sequences were submitted to the NCBI to obtain accession numbers for AludOBPs (ON419948-ON419978), AludCSPs (ON419979-ON419981), AludSNMPs (ON419982 and ON419983), AludORs (ON419984-ON420025), AludGRs (ON420026-ON420055), and AludIRs (ON420056-ON420076).

## Figures and Tables

**Figure 1 ijms-23-10531-f001:**
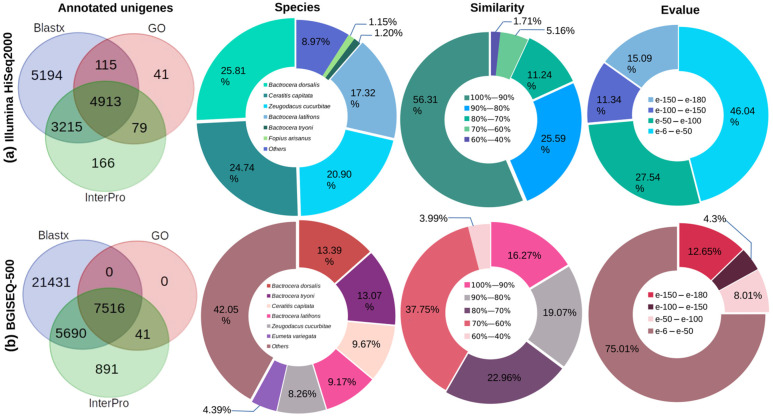
Functional annotation in GO, InterPro, and UniprotKB databases (accessed on October 2021) of unigenes obtained from *A. ludens* head transcriptomes constructed with Illumina HiSeq2000 (**a**) and BGISEQ-500 (**b**) platforms; also showing species distribution, similarity percentages, and E-value from BLASTx analysis against the Insecta-UniprotKB database.

**Figure 2 ijms-23-10531-f002:**
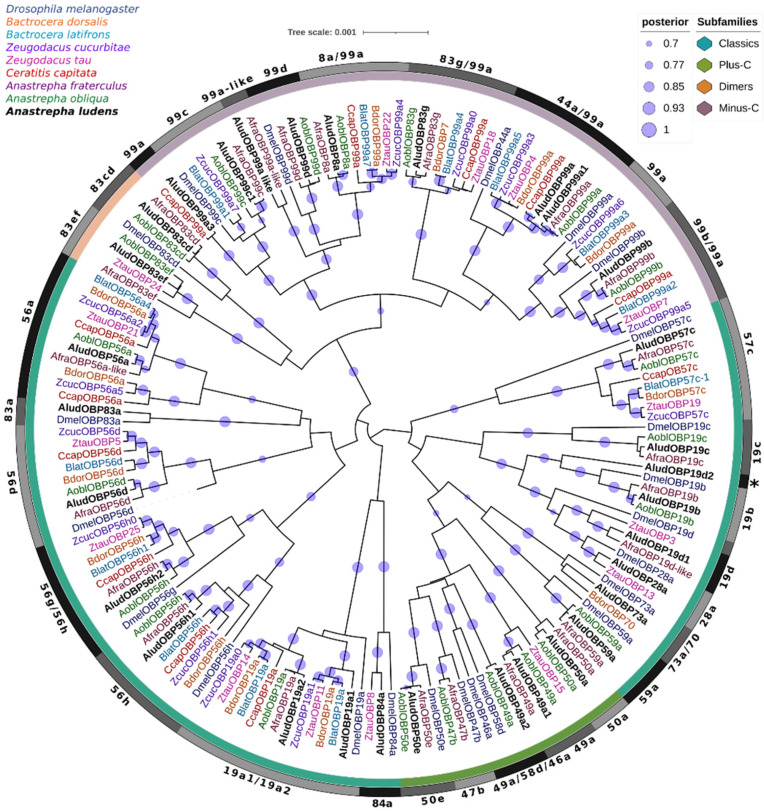
Bayesian phylogenetic analysis of candidate OBPs from *Anastrepha ludens* with homologs from seven Tephritidae species and the model organism *Drosophila melanogaster*. The clusters of subfamily OBPs identified in AludOBPs have distinct color bar marks. Gray bars delimit the branch expansions of the AludOBPs and their orthologues. Circles showed posterior probability values higher than 70%.

**Figure 3 ijms-23-10531-f003:**
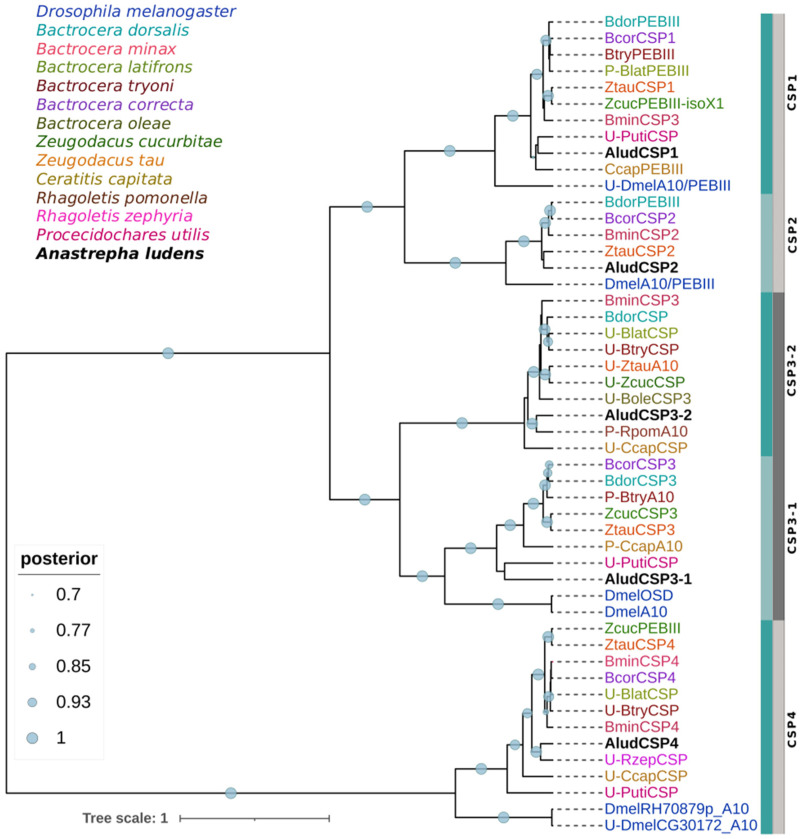
Bayesian phylogenetic analysis of candidate CSPs of *Anastrepha ludens* with homologs from twelve dipteran species and the model organism *Drosophila melanogaster*. Blue and gray bars delimit the branch expansions of the AludCSPs and their orthologues. Circles showed posterior probability values over 70%. Letters “U-” and “P-” refer to sequences in the databases that are uncharacterized or were automatically annotated by the genomic sequence prediction method.

**Figure 4 ijms-23-10531-f004:**
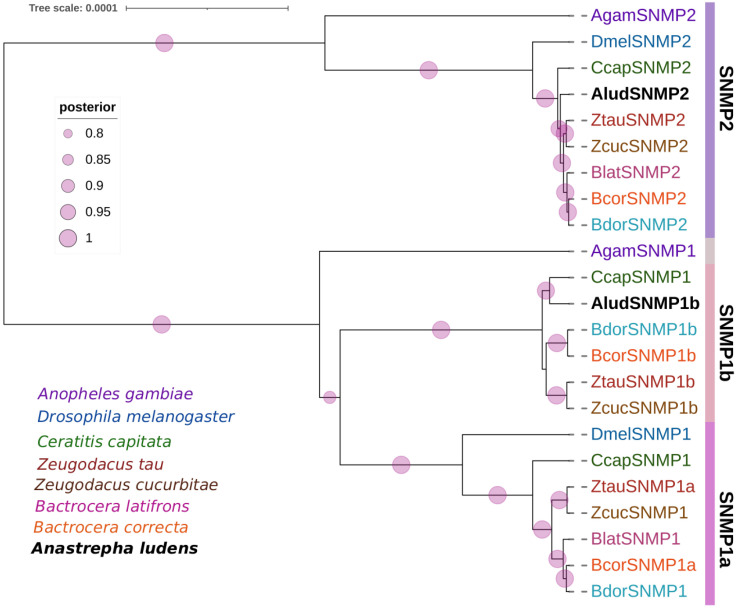
Bayesian phylogenetic analysis of candidate SNMPs from *Anastrepha ludens* with homologs from seven dipteran species and the model organism *Drosophila melanogaster*. Bars delimit clusters of AludSNMPs and their orthologues. Circles showed posterior probability values higher than 70%.

**Figure 5 ijms-23-10531-f005:**
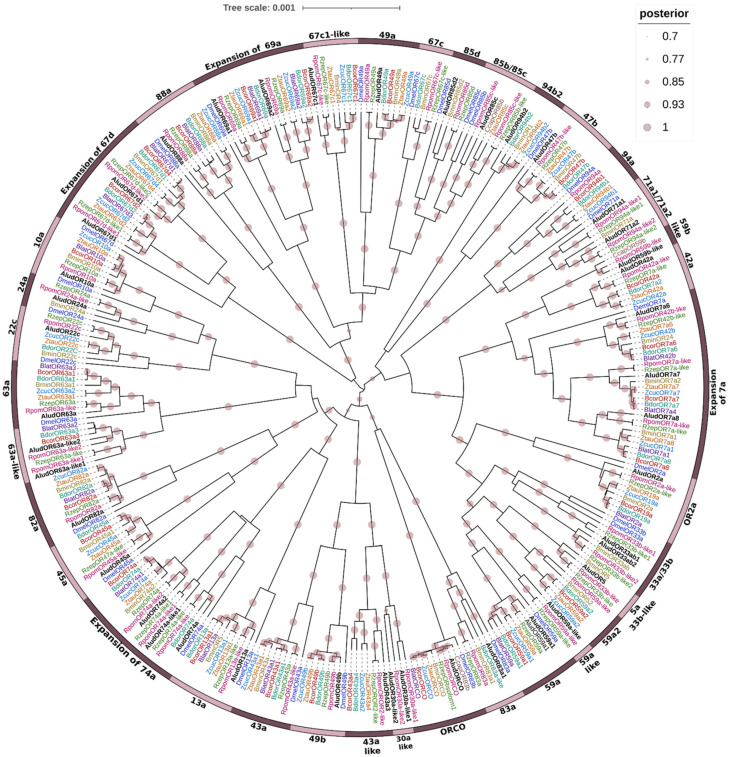
Bayesian phylogenetic analysis of candidate ORs of *Anastrepha ludens* with homologs from eight dipteran species and the model organism *Drosophila melanogaster*. Circles showed posterior probability values higher than 70%. Pink bars delimit clusters of AludORs and their orthologues.

**Figure 6 ijms-23-10531-f006:**
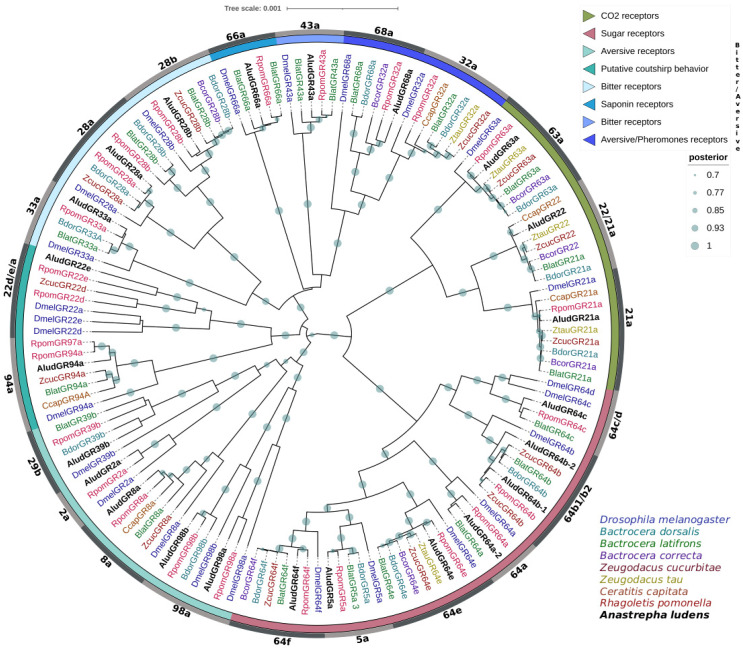
Bayesian phylogenetic analysis of candidate GRs from *Anastrepha ludens* with homologs from seven Tephritidae species and *Drosophila melanogaster*. Distinct colors represent clusters of AludGRs members based on putative functions reported in previous studies. Gray bars delimit branch expansions of AludGRs and their orthologues. Circles showed posterior probability values higher than 70%.

**Figure 7 ijms-23-10531-f007:**
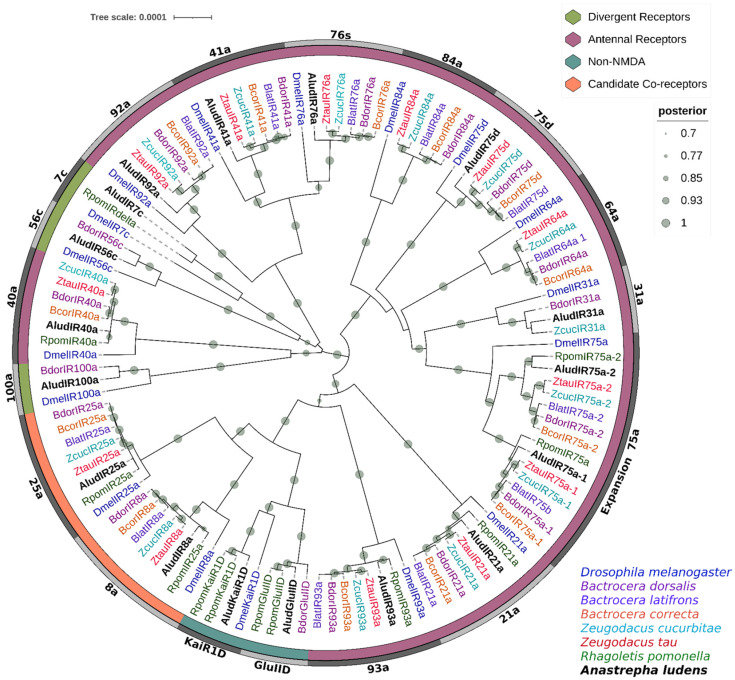
Bayesian phylogenetic analysis of candidate iGluRs/IRs from *Anastrepha ludens* with homologs from seven Tephritidae species and *Drosophila melanogaster*. Distinct colors represent clusters of AludIRs members based on putative functions reported in previous studies. Gray bars delimit the branch expansions of the AludIRs and their orthologues. Circles showed posterior probability values higher than 70%.

**Table 1 ijms-23-10531-t001:** Attributes of *A. ludens* Odorant-binding proteins subfamilies.

Subfamily	Number	Length (aa)	Protein Core Region	Similarity (%)
Classic	17	110–306	C1-X_36–171_-C2-X_3_-C3-X_32–50_-C4-X_8–21_-C5-X_8_-C6	26.41
Minus-C	8	102–173	C1-X_16–30_-C2-X_38–39_-C3-X_18–19_-C4 and C1-X_28_-C2-X_3_-C3-X_39–40_-C4-X_10_-C5-X_8_-C6	39.26
Plus-C	4	126–232	C1a-X_10–13_-C1b-C1c-X_13–19_-C1-X_9–48_-C2-X_3_-C3-X_43–44_-C4-X_19–33_-C5-X_9_-C6-X_8_-C7-X_10–11_-C8-X_9_-C9	28.87
Dimer	2	271–282	C1-X_28–34_-C2-X_3_-C3-X_31_-C4-X_10–14_-C5-X_8_-C6-X_17–28_-C1′-X_24–25_-C2′-X_3_-C3′-X_35_-C4′-X_17–21_-C5′-X_8_-C6′	35.65

## Data Availability

Not applicable.

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
