# Peer review of "Identification of Candidate Chemosensory Gene Families by Head Transcriptomes Analysis in the Mexican Fruit Fly, Anastrepha ludens Loew (Diptera: Tephritidae)"

_ijms, 2022, doi:10.3390/ijms231810531_

Round 1

Reviewer 1 Report

This work makes the first effort to identify, classify and characterize the six chemosensory gene families by analyzing two head transcriptomes of male and female adults of A. ludens. But it can not be accepted in current form. I  have some points that the authors may address to improve it. 

Comments are listed below:

Major points:

Some necessary qPCR experiments about six chemosensory genes family are needed.

It is incredible that only 3 CSPs were identified from Anastrepha ludens. I recommend the authors conduct iterative searching (using the 3 CSPs they identified) back into the transcriptome to identify more CSPs. The number of screened CSP is less than most of the published papers, therefore I suggest a carefully check on this point.

Are there any ODEs in your data?

Pheromone receptors and ORco belongs to a relatively conservative branch, which can be specifically pointed out in the evolutionary tree of OR. Moreover, ORco is an important subtype of OR. It is recommended to add a description in the results and discussion sections.

The number of these six chemosensory genes in the whole manuscript should be carefully checked. i.e. Line 27, 129, 134.

The number of species used for phylogenetic analysis were different among the six olfactory gene classes. Please carefully check it. And explain why you apply different species into phylogenetic analysis for different types. 

Minor points:

Line 27. Line 129. Total No. is not 154 chemosensory genes, please check it.

Line 47. “solubilizing” may not be the function of OBPs or CSPs.

Line 92, 98, 141, and so forth. Change “Supplementary Table T1” into Table S1 according to your supplementary file.

Line 109. Figure 1. (a) The centre of the pie chart was not clear (Evalue). Please also change a) and b) into (a) and (b) in the figure.

Line 134. “Of the 35 sequences” should be changed to “Of the 31 sequences.

Lines 158-159. “from six Tephritidae species and the model organism Drosophila melanogaster (Figure 2).” changed to “from seven Tephritidae species and…”.

Lines 212-213. Species should be italicized, check full text.

Line 247. Figure 4 may lay on line numbers. The authors should carefully read it before sbm

Line226, 246. Table 1 and Table 2 can be put in the supplementary.

Gene name should be italic and please check the whole manuscript.

Line 565-567. It is not clear that the age of samples used for RNA extractions to construct cDNA libraries. And why did the authors choose these ages?

Line 617-621. These sequences should be released before online publication.

Line 641. Raw data are usually included in this section. For example, they were submitted to NCBI for a project number.

The reference format should be checked.

Reviewer 2 Report

The authors, O.L. Segura León, B. Torres Huerta, A.R. Estrada Pérez; J. Cibrián Tovar; F. de la Cruz Hernández Hernández, J.L. Cruz Jaramillo, J.S. Meza Hernández, and F. Sánchez-Galicia, in their contribution “Identification of candidate chemosensory gene families by head transcriptomes analysis in the Mexican fruit fly, Anastrepha ludens Loew (Diptera: Tephritidae)” aim to identify, classify and characterize the six chemosensory gene families of the Mexican fruit fly Anastrepha ludens Loew, analyzing two head transcriptomes of male and female adults from breeding and wild populations.

This is a valuable contribution to the study of chemocommunication in Tephritidae. Given the high diversity of the olfactory system among the different groups of Diptera, investigations on species-specific insect olfactory system are welcome.

The standard of the manuscript is high. The information is very clear, well organised, and informative. The figures are relevant, the writing is correct, and the discussion is clear and comprehensive. The writing is agile and concise, and the quality of the figures is high.

 There are however three main concerns:

-    1) Should not the authors have done a validation of RNA-Seq data by qRT-PCR analysis, at least for a number of genes in each family? It would have done the RNA-Seq data more reliable. 

- 2) It would be very interesting having information about transcripts abundance. Not only to establish comparisons inside each group of genes, but also, if possible, to compare the different physiological stages.

   3) Line 535-537: “This work represents the first effort to identify, classify and functionally characterize the six chemosensory gene families from the analysis of head transcriptomes of male and female adults of A. ludens in different physiological stages.”

Although the authors include animals in different physiological stages, it is not sufficiently clear from the discussion what differences are found between the different stages studied. In fact, this relevant aspect is not addressed at all in either the results or the discussion section.

Minor issues:

-        The RIN values ranges should be indicated.

-        Line 41: “these perireceptor events involve the binding”

The term binding should be more specific.

Reviewer 3 Report

The manuscript titled "Identification of candidate chemosensory gene families by head transcriptomes analysis in the Mexican fruit fly, Anastrepha ludens Loew (Diptera: Tephritidae). The manuscript idea is good but I have some comments such as;

Abstract

Is written in good manner but it consists of

Background part from line 20-24

Materials and method part from line 25-26

Results from line 27-29

Convey from line 30-33  

I see that the background part in the abstract should be decreased and results part should be increased.  And the convey part should indicated to the aim of the study first and the importance of the obtained results second.

Introduction

1-Not contains a recent references especially 2020, 2021 and 2022 and this could be deficiency and not good for the manuscript.

2-did not contain the aim of the study or it is not quite clear and should be rewritten in good manner.

Results

I see that the authors used word "we" much, they not need to use this word as it is.

Discussion

It needs recent citation and more references especially in 2020, 2021 and 2022

It is too long and should be summarized

Materials and Methods

In line 545 4.1. Collection of biological material should be changed into Collection of biological material and identification

The insects should be identified in taxonomy lab based on scientific roles or using 18S rRNA by PCR.

CONCLUSION

A short listed conclusion should be added to the manuscript to explain the importance of the study.

Round 2

Reviewer 1 Report

I feel that the authors have fully addressed my questions. 

Reviewer 2 Report

The authors have made an effort to answer and reflect in the manuscript most of the issues raised by the reviewers and the result is a manuscript acceptable for publication.